# A Survey on Biosecurity and Animal Welfare in Twenty-Five Beef Cattle Farms in the Marche Region, Central Italy: Application of the ClassyFarm Checklists

**DOI:** 10.3390/ani15030312

**Published:** 2025-01-23

**Authors:** Filippo Alquati, Martina Quagliardi, Alessandra Gavazza, Alessandra Roncarati, Livio Galosi, Corrado M. Corradini

**Affiliations:** School of Biosciences and Veterinary Medicine, University of Camerino, Via Circonvallazione 93/95, 62024 Matelica, MC, Italy; filippo.alquati88@gmail.com (F.A.); alessandra.gavazza@unicam.it (A.G.); alessandra.roncarati@unicam.it (A.R.); livio.galosi@unicam.it (L.G.); vet.corradini@gmail.com (C.M.C.)

**Keywords:** ClassyFarm platform, beef cattle farm, animal welfare, biosecurity, checklist

## Abstract

In the Italian livestock scenario, the ClassyFarm platform (CFp) has been activated to help farmers, farm veterinarians, and public veterinary services to assess livestock farms’ biosecurity and welfare through a risk evaluation. Data for self-monitoring purposes can be added into the platform. This study aimed to analyse the ClassyFarm rating system’s application in the Marche Region, Central Italy. Traditionally, this region is characterised by both intensive and traditional/extensive breeding, often with familial management. Seven fattening and eighteen cow–calf line farms (a total of twenty-five farms) were taken into consideration. Utilising checklists of the CFp, these farms were visited and evaluated. All the livestock farms received the lowest rating for biosecurity, in line with the national average, and the highest scores for animal welfare. This project highlighted the peculiarities of the rearing systems in the Marche Region, valorising the usefulness of the CFp in the regional livestock scenario.

## 1. Introduction

In Italy, since 2017, the ClassyFarm platform (CFp) has been activated and developed to assess and categorise livestock farms according to the risk evaluation. This online platform was created by the Experimental Zooprophylactic Institute of Lombardy and Emilia-Romagna (IZSLER), at the request of the Italian Ministry of Health, in order to adhere to the objectives of Regulation (UE) 2016/429 (Animal Health Law). This regulation aims at preventing and controlling animal diseases that can be transmitted to other animals or humans. A higher awareness from farmers and vets, who are involved in early detection to prevent major disease outbreaks, is among the key priorities in tackling diseases [1]. The system represents an informatic tool useful for assessing the status of an Italian farm regarding the risk of disease by means of data collection concerning animal production, management, biosecurity, structures, health status, and the use of drugs. IZSLER works as the administrator of the system, which is managed through the portal “Vetinfo”. It was created by the Ministry of Health, with the aim of collecting and presenting both health- and non-health-related data, useful for the management of the national Animal Health and Food Safety system. Particular attention is paid to the definition of health risks along the entire production chain, from the production of animal feed to the marketing of food for human consumption. It provides a series of public services such as news, references to current legislation, information of interest, statistical data, zootechnical and pet registers, and a veterinary drug prescription system [2]. The CFp was developed with the aim to improve collaboration between competent authorities, veterinary practitioners, and livestock breeders, also through self-monitoring, to guarantee to all people in the food chain, from farmers to consumers, the highest levels of animal health [3] and the safety and quality of livestock products [4]. The registration of an Italian livestock farm in the CFp is on a voluntary basis; however, it is a compulsory prerequisite to access the CAP (Common Agricultural Policy) [5], as this system was developed alongside the livestock sanitary regulations and is integrated via a questionnaire (checklist) used by Veterinary Public Services in their livestock control activities [6]. Checklists for most of the livestock species and several breeding systems have been drawn up using expert knowledge to support official control, gather information, and encourage the application of welfare and biosecurity levels. To access and use these checklists, the veterinarians have to be trained by personnel from IZSLER [7]. Due to the CFp, it is possible to assess livestock farms individually by following handbooks and checklists, with the possibility of comparing these assessments to the results of other similar farms on the local, regional, or national level [8]. In the case of beef rearing, the checklist is organised into two thematic areas concerning “Company management and personnel” (Area A) and “Structures and equipment” and “Major risks and alarm systems” (Area B). The parameters related to the analysis of the presence or absence of detectable adverse effects on animals are conveyed by the information system into another area (Area C), which reports the results of the main “animal-based measures”, such as animal behaviour, the degree of cleanliness, lameness, and skin lesions. The scores are evaluated according to a qualitative scale from “optimal” to “insufficient” with intermediate levels (“good”, “sufficient”) assigned by the operator and elaborated by an algorithm that provides a percentage for comparison with the national situation and other farms. The aim is to provide a constant snapshot of a farm for the better protection of public health. In this way, the operator can identify the critical points and areas for improvement of their farm, with the possibility of implementing effective measures to reduce the level of risk and improve their activity. Furthermore, the system is structured in such a way as to make it possible to compare the individual farm with similar farms in terms of production at regional and national level. Many consortia and associations of consumers impose adhesion to CFp from the producers to increase the attention to animal health and apply the One Health approach.

Animal welfare is a crucial parameter in this system, and animals’ behavioural indicators, such the activities of walking, standing, or feeding, play a significant role in its determination [9]. In addition, production parameters may also be considered for the evaluation of animal welfare, since the association between these factors has been documented in livestock holdings [10]. 

Biosecurity has a crucial role in not only protecting animals from disease but also significantly limiting the overuse of antimicrobials and antimicrobial resistance, as reported for farms in the north of Italy [11,12]. At the same time, as ascertained by other studies [3,9,13], appropriate biosecurity measures are fundamental. Suitable biosecurity measures also ensure efficient productive performance and high profitability for the farmer, reducing the costs due to animal care and avoiding production losses [14]. The risk of the entry and spread of possible pathogens cannot be definitively eliminated, but the contribution to reducing the frequency of diseases in the herd and their severity must become a fundamental priority [15].

In Italy, in terms of numeric consistency, a recent study by Tamba et al. [16] classified Italian cattle farms into two types, considering consistency and geographic location: the intensive cattle farming in the northern area of the country and the extensive cattle farming in the middle and southern parts of the country. In particular, in the areas of the northwest, calves of local breeds are mainly reared, especially the autochthonous Piemontese breed; while in the northeast, the breeds of cattle raised come from other European countries, usually of French origin [17].

The CFp has been created to be integrated into the system of veterinary epidemiological surveillance networks to guarantee the exchange of information between the food business operator, who breeds animals for food production, and the competent authority, such as the Veterinary Services. In northern Italy, the application of the CFp was studied in South Tyrol province (Trentino Alto Adige region), where biosecurity practices were evaluated in small-scale mountain dairy farming [18,19,20], in the Valle d’Aosta region, where there was tie-stall rearing [21], and more in the centre of the peninsula, in the Tuscany region, where different beef cattle farms were evaluated [10]. Moreover, the application of the CFp was studied in other species, such as dairy sheep flocks in Tuscany [22], assessing the hygiene of buffalo milk parlours in the Campania region [23] and assessing the air quality of pig farming [24].

In the centre and south of the Italian peninsula, beef cattle breeding comprises farming systems related to a traditional rural farming approach [16]. The significance of the local meat production, the diversity of meat products, and its rich cultural and culinary heritage centred around meat production represent important socioeconomic traits. In the Marche Region, cattle breeding has undergone a continuous reduction, which has also affected the local Marchigiana breed, especially after the earthquake in year 2016. After that dramatic event, population losses were recorded in the seismic crater, including Macerata province, with negative effects for agriculture and zootechnics [25,26,27]. The rearing technique is represented by the calf–cow line, in both traditional/extensive and intensive breeding [28], on small- or medium-sized farms [29]. Beef production is based on a cow–calf system that considers a different management approach in relation to the physiological phase, as recognised from a zootechnical point of view at an international level [30]. Macerata province is renowned for its local meat production, particularly cattle and pig breeding. The area is known for its prized “Marche breed” of cattle, which is esteemed for the quality of its meat.

Based on this scenario, to improve the productivity of beef cattle farms in Central Italy, this study was performed on beef cattle farms located in the Marche Region in order to verify the application and usefulness of the CFp with particular attention to animal welfare and biosecurity, and the results were compared to the national data.

## 2. Materials and Methods

Twenty-five beef cattle farms (1–25 F) located in the Marche Region, Central Italy, were considered during the years 2021–2023 for this survey (Figure 1), using the indications and the checklists provided by the ClassyFarm platform (CFp, www.classyfarm.it, accessed on 12 October 2024).

Data regarding the numerical consistency and categories of the Marchigiana breed reared in the farms were collected from each farm. The CFp checklists evaluate several parameters to classify the biosecurity and animal welfare level of each farm.

Table 1 and Table 2 show the main parameters considered for the assessment of the management practices regarding biosecurity and animal welfare, respectively. These parameters are included in Area C of the questionnaire available for the operators. The number of parameters used for the animal welfare assessment was different for the fattening (56) and cow–calf line (72) farms. This dissimilarity is due to the fact that some control points on the animal welfare checklist refer only to specific animal categories. When the collection of data was completed, scores for both biosecurity and animal welfare were assigned to the farm [31,32].

To assess each parameter, the rating consisted of “poor”, “acceptable”, or “excellent”.

The biosecurity assessment consisted of 15 parameters indicated in the checklist, divided by the authors into three groups to properly represent them: “low”, from 0 to 5 parameters; “medium”, from 6 to 10; and “high”, from 11 to 15. The same parameters were used for both the fattening and cow–calf line farms.

The same approach was applied for the animal welfare assessment, differentiating between the fattening and cow–calf line farms, as the first considers 56 parameters while the second considers 72 parameters. In this case, the ratings were divided by the authors into four levels: “low”, “low–medium”, “medium–high”, and “high” (Table 3).

GraphPad Prism 10.4.0 (621) was used to graphically represent the data. Bar graphs were used to illustrate how the farms were distributed among the ratings. Floating bars were used to represent the total score of the farms in terms of percentages, comparing them with the national averages provided by the CFp. The biosecurity and overall animal welfare (OAW) weighted averages were also calculated by the platform, assigning a different weight to each area of interest. The percentage distributions of the biosecurity and animal welfare assessments were assessed using Shapiro–Wilk normality test. Consequently, the means were compared using Welch’s *t*-test, since the numerosity of the two groups was different, using the software R Version 4.4.2. The data were considered significant with *p* value < 0.05.

## 3. Results

For each farm, all the evaluated parameters and the ratings given and uploaded on the CFp can be found in the excel file in the Appendix A.

In Figure 2, the numerical consistency and categories of animals found in the twenty-five farms are described.

In Figure 3, concerning the biosecurity ratings, most of the fattening farms received “acceptable” ratings for all of the evaluated parameters. A “poor” rating was assigned to 1-F and 7-F. An “excellent” rating for the farms was used very rarely.

The cow–calf line farms showed a conspicuous number of “acceptable” rates. Even here, two farms (22-F and 23-F) received a “poor” rating, although this score was very rarely used. Finally, an “excellent” rating was rarely used for any of the assessed farms.

As shown in Figure 4, the fattening farms had an “acceptable” score with medium–high frequency for most of the farms; only 1-F was in the low–medium range. The “poor” rating was scarcely used in this area; instead, an “excellent” rating was given with low–medium frequency.

The same distribution of “acceptable” ratings is found in Figure 4b; 21-F reached the highest frequency of this score; conversely, it was given to the other farms with a medium–high frequency. In the cow–calf line farms, a “poor” rating was not frequently given. At the same time, an “excellent” rating was given to most of the farms at a low–medium frequency, and only seven farms had a low frequency.

Based on the assessment performed, a total of 76 low ratings were obtained in the biosecurity area for all the farms, while for the animal welfare area, these low ratings totalled 99.

These results for the assessed farms were also calculated in terms of percentages and the Shapiro–Wilk test, except for the large hazard area, showing that the data were normally distributed.

In Figure 5, for the fattening farms, four (1-F, 3-F, 6-F, and 7-F) had values lower than the national average (52%), while the others had slightly higher values (60% and 63%). For the cow–calf line farms, only 11-F and 23-F were below the national average (42%). These data clearly show that at a national level, biosecurity has a broad range for improvement, as well as for the assessed farms. However, the differences between the means of the two breeding systems turned out not to be significant (*p* = 0.19).

In Figure 6, the percentages produced by the animal welfare assessment for the different areas and the national averages are illustrated. As shown in Figure 6a, the fattening farms obtained the highest rate in Area C and the lowest rate in large hazard areas, especially due to the scores of 1-F and 2-F. Overall, the farms showed very similar percentages compared to the national average, and the animal welfare rating was much higher than that of biosecurity.

As shown in Figure 6b, the highest value for Area C was registered in the cow–calf line farms. Farms 18-F and 19-F showed values below the national average. Indeed, a significant statistical difference was found between the two breeding systems concerning this area (*p* = 0.03), with the cow–calf line presenting the lowest value. Here, there was a large discrepancy in the mean value of Area B, with 14 farms showing lower percentages than the national reference average. In addition, the large hazard area also presented a very wide range of results, and only two farms (9-F and 24-F) had the lowest values. Overall, the farms exhibited higher values for the animal welfare area and were similar to the national reference average compared to the biosecurity area. Therefore, for the OAW, the cow–calf line percentage turned out to be significantly lower compared to the fattening breeding system (*p* = 0.02).

## 4. Discussion

Based on the results obtained at the end of the study, the application of ClassyFarm checklists permitted the identification of positive and negative elements of the biosecurity and animal welfare assessment, providing results in line with those of other studies performed in the northern part of the country, although these were performed on dairy cattle farms [19,33]. A recent study [34] showed that different strategies are needed to counteract antimicrobial resistance in animal husbandry.

In the present study, the results of the ClassyFarm checklists for the management practices of the twenty-five beef cattle farms reflected a family-based and rural approach due to the small scale of the farms, especially compared to the those in northern Italy [35]. Among these strategies, biosecurity can play an important role, and a series of management practices can be employed to identify the critical control points of a livestock farm [14]. Biosecurity is a measure that was recently introduced in animal production. As reported in a commentary in year 2022 [36], reviewing the evolution of studies dedicated to biosecurity, the use of this term arose in the early 2000s. At the international level, the World Health Organization introduced the role of biosecurity, and in Europe, as mentioned in the introduction, the “Animal Health Law” Regulation of 2016 contemplated the importance of measures to protect farms from pathogens, in order to prevent the use of antimicrobials, with the aim to adopt the “One Health” perspective [1].

Since then, in Italy, efforts have been conducted to promote the application of a multidisciplinary approach to address health risks originating from the environment–animal– human ecosystem interface.

Based on the results that emerged from the current study, the incidence of low ratings collected in the biosecurity assessment showed a much greater impact than those in the animal welfare assessment on the farm’s risk. This situation is presumably due to the small-scale enterprises of beef cattle, associated with the distance between the stables and a low density of beef cattle farm per ha [36,37], as showed in a survey on biosecurity and management practices in Belgian cattle farms in which external and internal biosecurity measures were analysed [38]. In this reference, it emerged that few actions were undertaken with a consequent increase in risks of pathology transmissions within and between herds [38]. The small distances among neighbouring farms were considered a serious factor that could facilitate the entrance of disease agents within the farm [38]. In the same study, exploring the biosecurity according to the herd size, no significant differences were found, and the group of farms with the lowest biosecurity level was represented by mixed-species farms [38].

In the current survey, the analysed farms started their activities many decades ago, and their management did not include biosecurity measures. The “small-is-beautiful” theory is still widely held in the public consciousness, supporting the idea that small farms are better for environmental preservation, animal welfare, and product quality, although less profitable [39]. Nowadays, the adoption of fences, barriers, and regulations for visitors’ entrance represents only some of the examples of the efforts the farmers must make to apply biosecurity measures [37].

This situation is in line with the main findings of biosecurity studies in Belgian cattle farms where most of the farmers involved considered their biosecurity level satisfactory, even if the overall level of biosecurity of their farms was assessed as low [36]. In Finland, although the sanitary status was assessed as good, the authors argued that it was hard to convince the farmers to improve the biosecurity level when the risks threatening their production were underestimated [40].

In the current study, the results showed that all the farmers paid particular attention to the introduction of new animals into the farm as part of good management practices and the possibility of correlating the improvement in herd health management with better production. The adoption of guidelines concerning the arrival, transport, and handling of new animals could help the farmers to prevent any unnecessary distress to the specimens. Our results showed the importance of hosting the animals in a quarantine area dedicated to animals after their arrival at the destination and are in agreement with previous studies on the effects of transportation on cattle health [41,42,43].

Moreover, the farmers that work on small-scale farms can be more familiar with the individual animals, and they can easily report on a daily basis when critical and pathological statuses are highlighted. This result was presumably possible thanks to the adhesion to the CFp. This platform allows strong collaboration between farmer and veterinarian in beef cattle management and can fill the gaps perceived between smaller and larger farms. In the lively debate on farm size [38,44,45] and its impact on animal welfare and the consistency of reared animals, larger farms have more skills available to perform early and more effective diagnosis on cattle compared to smaller farms in which no technical manpower is permanently employed to perform routine activities [46]. Thus, in the area in which the present survey was carried out, the CFp also provides opportunities to improve the professionalism and updating of the smallholder. There is a need to improve access to information and implement farmers’ knowledge, as confirmed by studies performed both in developing countries, where animal farming helps against malnutrition, and in rural areas of developed countries, as in Central Italy, where family management is widespread [36,47,48]. However, high animal welfare was achieved in the farms where farmers had positive engagement with both the agricultural and non-agricultural community and were satisfied with their land organisation [49].

The results concerning live animals’ handling in chutes, diseases and sanitary monitoring, and the disinfection of trucks at the entrance showed a biosecurity situation that must be improved, since there were many poor scores.

Regarding animal welfare assessment, the fattening farms showed better values than those of the cow–calf line farms, where both Area C and the OAW showed a mean that was significantly lower than that observed for fattening. This difference could be attributed to the weaning phase, which could affect the onset of health problems in calves during the transition to a solid diet when separation from the dam occurs. This period is recognised as a critical phase, as it is when the young stays with the mother for months [50]. A recent review described the multiple stressors affecting the calves in the first part of life, including the pre-weaning time [51]. Possible management deficiencies may cause health problems that occur at the end of weaning [52]. In Europe, to better evaluate animal welfare conditions, animal-based measures [53] have been proposed, and new indicators are now being considered, specifically regarding the welfare of calves and their weaning phase [54,55]. Concerning this study, the difference in farm numerosity between the two rearing systems could have very likely affected the results, although the evaluations were made by the same veterinarian.

## 5. Conclusions

This work provided additional information about the ClassyFarm platform, a system that generated a reliable overview of each beef cattle farm in the Marche Region. The results concerning the animal welfare assessment, which were above the national average, are probably due to the better general awareness and management concerning the aspects relating to animal welfare; however, they also highlighted the lower consideration for biosecurity in terms of livestock. The accurate analysis of these results should stimulate farmers and veterinarians to clearly identify critical points and, consequently, improve farming management. Based on this study, training of the farmers on the importance of biosecurity must be considered as a strategy to monitor and control the critical risk factors of productive performance. Further studies are needed to analyse all the potential uses of the platform in extensive cattle farming to improve both animal welfare and biosecurity.

## Figures and Tables

**Figure 1 animals-15-00312-f001:**
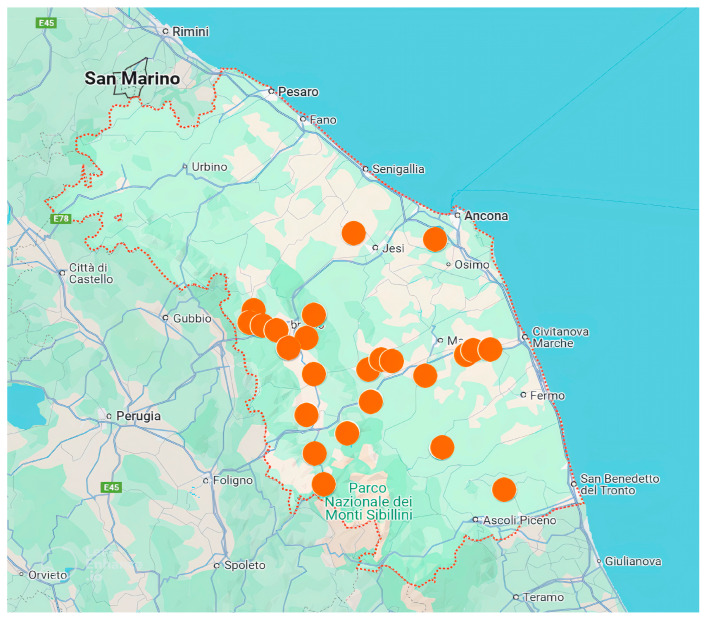
Territory of the Marche Region (red dotted line) and the twenty-five farms used for the study (orange points). Map data© 2022 Google.

**Figure 2 animals-15-00312-f002:**
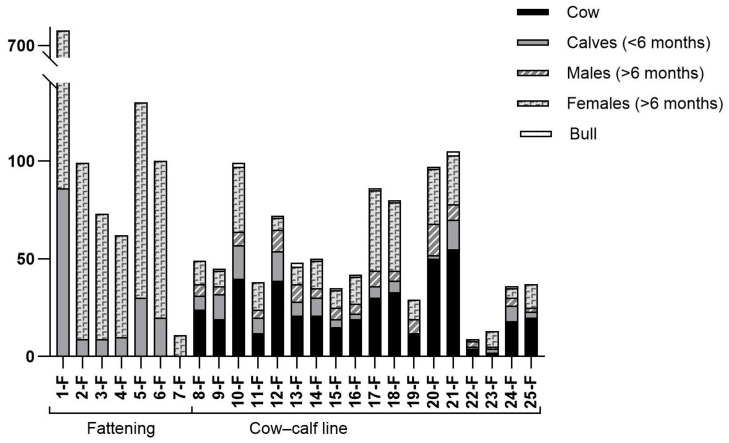
Numerical consistency and categories of animals found in the selected farms in the Marche Region, Central Italy.

**Figure 3 animals-15-00312-f003:**
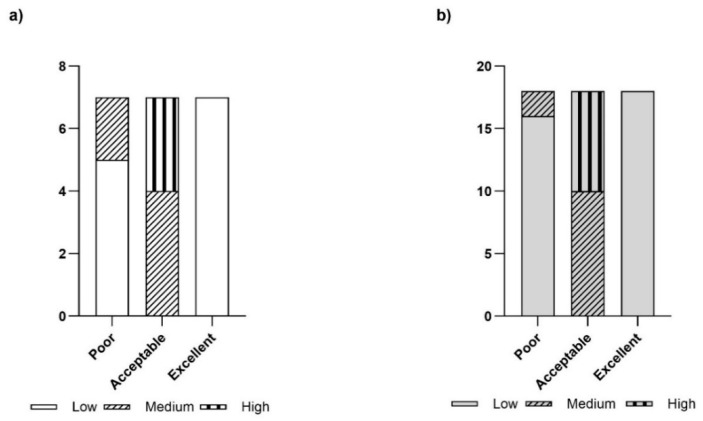
Results of the overall biosecurity parameters ratings in the assessed farms: (**a**) fattening farms, *n* = 7; (**b**) cow–calf line farms, *n* = 18.

**Figure 4 animals-15-00312-f004:**
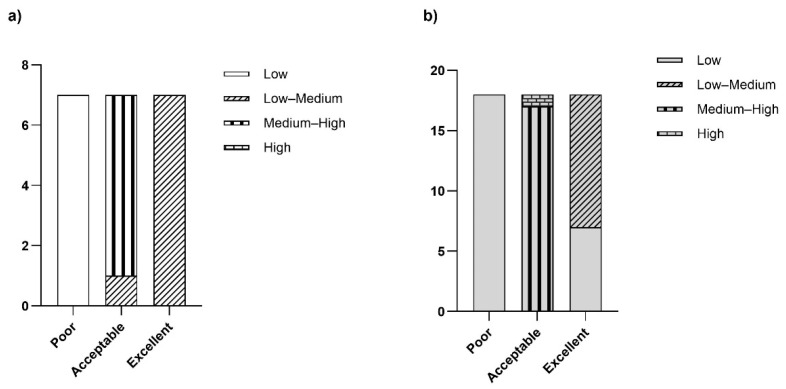
Results of the overall OAW ratings in the assessed farms: (**a**) fattening farms, *n* = 7; (**b**) cow–calf line farms, *n* = 18.

**Figure 5 animals-15-00312-f005:**
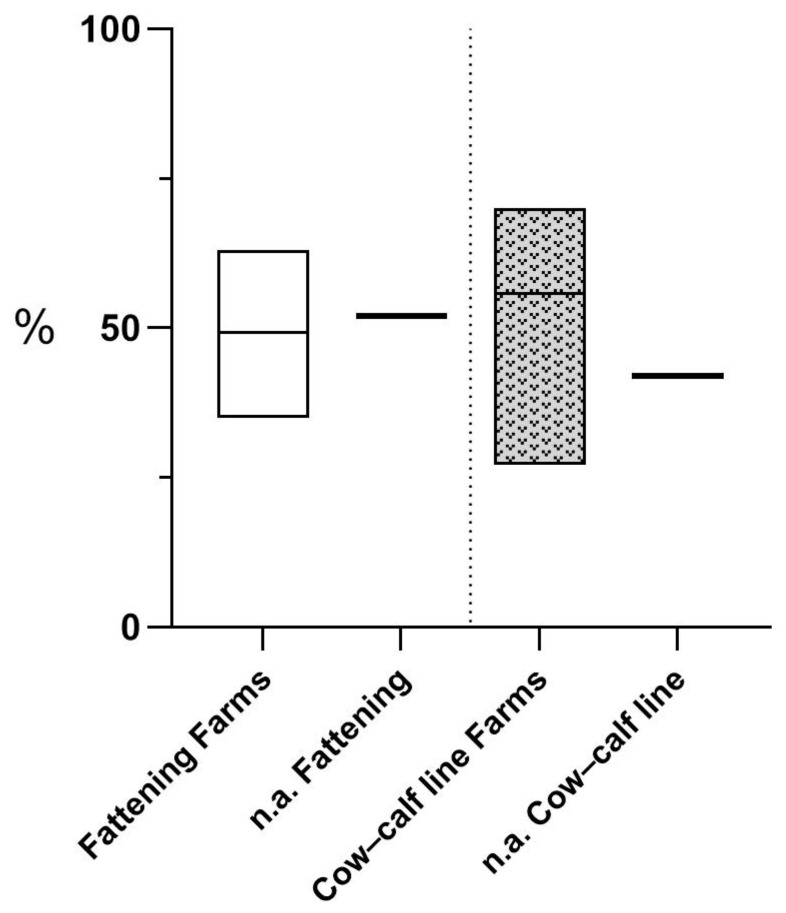
Comparison between the results obtained for the overall biosecurity assessment within the assessed farms and the national average (n.a.).

**Figure 6 animals-15-00312-f006:**
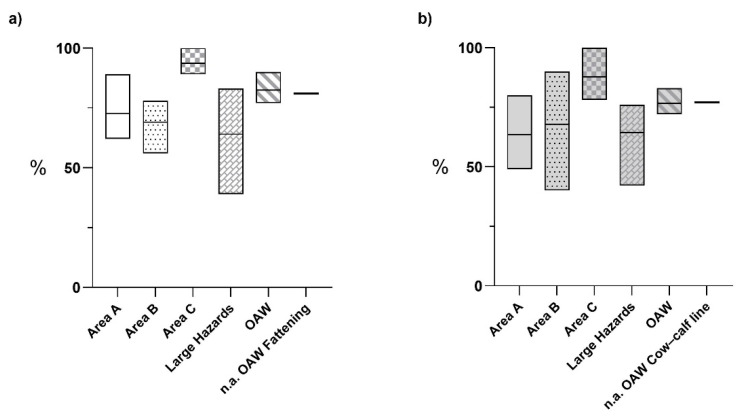
Comparison of the results obtained for the overall animal welfare (OAW) assessment within the assessed farms and the reference national averages (n.a.). (**a**) fattening farms, *n* = 7; (**b**) cow–calf line farms, *n* = 18.

**Table 1 animals-15-00312-t001:** Main items in the biosecurity assessment of the CFp.

Biosecurity Assessment
Disinfection system for vehicles
Loading live animals
Purchase and/or movement of animals off-farm
Quarantine/access management
Control and prevention of major infectious diseases
Health monitoring activities
Possibility of contact between foreign vehicles and reared animals

**Table 2 animals-15-00312-t002:** Main items in the animal welfare assessment of the CFp.

Animal Welfare Assessment			
Area A	Area B	Area C	Area’s Large Hazards and Warning Systems
Corporate and Personnel Management	Facilities and Equipment	Animal-Based Measures	
Number of staff	Outdoor shelters	Agonistic behaviours test	Origin of the water
Animal grouping strategy	Housing system	Avoidance distance test	Noise
Daily inspections on animals	Type of flooring	Body condition score	Lighting for inspections
Culling	Facilities for sick animals	Animal cleanliness	Ventilation system alarm
Animal handling	Lighting	Respiratory symptoms	Fire alarm
Prevention of neonatal diseases	Equipment	Mortality rate	Register of drug treatments

**Table 3 animals-15-00312-t003:** Classification of biosecurity and animal welfare parameters.

	Low	Medium	High	
Biosecurity	0–5	6–10	11–15	
	Low	Low–Medium	Medium–High	High
Animal Welfare (Fattening Farm)	0–14	15–28	29–42	43–56
Animal Welfare (Cow–Calf Line Farm)	0–18	19–36	37–54	55–72

## Data Availability

Data available on request from the authors.

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
