# Peer review of "A Survey on Biosecurity and Animal Welfare in Twenty-Five Beef Cattle Farms in the Marche Region, Central Italy: Application of the ClassyFarm Checklists"

_animals, 2025, doi:10.3390/ani15030312_

Round 1

Reviewer 1 Report

Comments and Suggestions for Authors

In this study, Alquati et al. analyzed the ClassyFarm (CFp) rating system's applicability in the Marche Region, Central Italy. This work aimed to analyse the applicability of CFp in the biosecurity and animal welfare areas in twenty-five beef cattle farms of the Marche Region, central Italy, using the checklists provided by the platform itself. The results reflects the type of breeding typical of this region, characterized by particular attention to animal welfare, guaranteed by non-intensive and family-run breeding. The design of this study is scientifically sound, and the data is credible. However, some issues need to be addressed before the manuscript can be published.

1. The abstract lacks quantifiable conclusions; the author needs to strengthen the quantitative results and enhance the practical significance of the findings.

2. The caption should be at the bottom of the image, not at the top. Please modify it.

3. The statistical analysis is missing in Figure 5; please add the statistical analysis and revise the corresponding text description.

4. The description of the results in the manuscript needs to be revised, especially to avoid the phrase "Figure shows.

Author Response

Comments and Suggestions for Authors

 In this study, Alquati et al. analyzed the ClassyFarm (CFp) rating system's applicability in the Marche Region, Central Italy. This work aimed to analyse the applicability of CFp in the biosecurity and animal welfare areas in twenty-five beef cattle farms of the Marche Region, central Italy, using the checklists provided by the platform itself. The results reflects the type of breeding typical of this region, characterized by particular attention to animal welfare, guaranteed by non-intensive and family-run breeding. The design of this study is scientifically sound, and the data is credible. However, some issues need to be addressed before the manuscript can be published.

  1. The abstract lacks quantifiable conclusions; the author needs to strengthen the quantitative results and enhance the practical significance of the findings.

AU: done.

  1. The caption should be at the bottom of the image, not at the top. Please modify it.

AU: done.

  1. The statistical analysis is missing in Figure 5; please add the statistical analysis and revise the corresponding text description.

AU: we are not running any statistical test, since the results are just descriptive.

  1. The description of the results in the manuscript needs to be revised, especially to avoid the phrase "Figure shows.

AU: done.

We used the MDPI English editing service.

Reviewer 2 Report

Comments and Suggestions for Authors

This manuscript aims to analyze the applicability of CFp in 25 beef cattle farms in the Marche Region, central Italy. CFp has been activated to assess livestock farms' biosecurity and welfare and provides information for consumers. All livestock farms received the lowest rating for animal biosecurity, even if in line with the national average and the highest scores for animal welfare. This result reflects the breeding typical of this region, characterized by particular attention to animal welfare, guaranteed by non-intensive and family-run breeding. The study is intriguing; however the language needs a thorough revision, and the content requires further improvements. For example, in the methods part, more information needs to be provided, including who was responsible for the assessment, whether the same group of people assessed all the farms, the date of the assessment, and so on. In the results section, the author's description is confusing. In addition, the discussion section is not sufficiently in-depth. It would be helpful to specifically analyze factors contributing to the low biosecurity scores and high animal welfare scores. Most importantly, the discussion of the usefulness of the CFp in the beef cattle farm of Marche region is limited. This paper has the potential to be published after thorough revision.

Introduction:

1.       Please revise this sentence to make it more understandable. This regulationhazardous agents. (Line44-46)

2.       More detailed description of the operation and evaluation metrics of CFp should be provided.

Method:

1.       The detailed parameters of CFp used in the study and assessment of each farms should be provided at least as supplementary materials.

2.       The rationale of making the cutoff poor, acceptable, excellent and “low, medium, high” should be provided. (Line 114-123)

3.       The aim of this study is to analyze the applicability of CFp in biosecurity and animal welfare areas. However, the method fails to evaluate the usefulness of CFp in biosecurity and animal welfare. More analysis needs to be added.

Results

1.       The depiction of Figures 2 and 3 is difficult to understand. (Line 136-139, 151-155)

2.       The statistical methods used for data analysis need to be explained, as well as how the rating results are derived from the raw data.

3.       The results section should clearly display the ratings for each farm, as well as comparisons with the national average. Charts should be clear, accurate, and easy to understand. The p value should be provided.

4.       The word cow-calf fine farms should be cow-calf line farms. (Line 144,160)

Discussion

1.       It’s interesting that the beef cattle farm in the Marche region have good animal welfare while low biosecurity score. It would be a good discussion point if the manuscript could dive deep into these results. Analyzing the specific reasons for these results would be a great resource for improvements of these farms. Again, the discussion fails to evaluate the applicability of the CFp in Marche region.

2.       This paragraph looks more suitable for introduction instead of discussion. (Line 229-238).  

References

1.       The format of the reference should be unified.

Comments on the Quality of English Language

The English expression throughout the manuscript is very confusing, and the content in the introduction and discussion sections is difficult to understand. The methodology, discussion, and conclusion do not adequately support the purpose of the study. The topic is interesting, but the paper requires a comprehensive revision.

Author Response

Comments and Suggestions for Authors

This manuscript aims to analyze the applicability of CFp in 25 beef cattle farms in the Marche Region, central Italy. CFp has been activated to assess livestock farms' biosecurity and welfare and provides information for consumers. All livestock farms received the lowest rating for animal biosecurity, even if in line with the national average and the highest scores for animal welfare. This result reflects the breeding typical of this region, characterized by particular attention to animal welfare, guaranteed by non-intensive and family-run breeding. The study is intriguing; however the language needs a thorough revision, and the content requires further improvements. For example, in the methods part, more information needs to be provided, including who was responsible for the assessment, whether the same group of people assessed all the farms, the date of the assessment, and so on. In the results section, the author's description is confusing. In addition, the discussion section is not sufficiently in-depth. It would be helpful to specifically analyze factors contributing to the low biosecurity scores and high animal welfare scores. Most importantly, the discussion of the usefulness of the CFp in the beef cattle farm of Marche region is limited. This paper has the potential to be published after thorough revision.

Introduction:

  1. Please revise this sentence to make it more understandable. “This regulation…hazardous agents.” (Line44-46)

AU: done (lines 45-48)

  1. More detailed description of the operation and evaluation metrics of CFp should be provided.

AU: done

Method:

  1. The detailed parameters of CFp used in the study and assessment of each farms should be provided at least as supplementary materials.

AU: we added it in supplementary materials.

  1. The rationale of making the cutoff “poor, acceptable, excellent” and “low, medium, high” should be provided. (Line 114-123)

AU: done (lines 163-166).

  1. The aim of this study is to analyze the applicability of CFp in biosecurity and animal welfare areas. However, the method fails to evaluate the usefulness of CFp in biosecurity and animal welfare. More analysis needs to be added.

AU: the aim of the study was reformulated taking into account the scope of the actions performed.

Results

  1. The depiction of Figures 2 and 3 is difficult to understand. (Line 136-139, 151-155)

AU: we modified the descriptions (lines 184-186; 197-201).

  1. The statistical methods used for data analysis need to be explained, as well as how the rating results are derived from the raw data.

AU: we didn’t use any statistical test, since they are descriptive data.

  1. The results section should clearly display the ratings for each farm, as well as comparisons with the national average. Charts should be clear, accurate, and easy to understand. The p value should be provided.

AU: it’s not possible to show all the results of the ratings of each farm otherwise the representation will become too big. The results can be found in the supplementary materials.

  1. The word “cow-calf fine farms” should be “cow-calf line farms”. (Line 144,160)

AU: done (lines 191, 206-207).

Discussion

  1. It’s interesting that the beef cattle farm in the Marche region have good animal welfare while low biosecurity score. It would be a good discussion point if the manuscript could dive deep into these results. Analyzing the specific reasons for these results would be a great resource for improvements of these farms. Again, the discussion fails to evaluate the applicability of the CFp in Marche region.

AU: we implemented the Discussion part concerning good animal welfare and low biosecurity scores. We changed the general meaning of “applicability” into “application” of the ClassyFarm system because it is much adherent to the work we did.

  1. This paragraph looks more suitable for introduction instead of discussion. (Line 229-238).

AU: The paragraph in object has been moved to introduction and revised (Lines 96-104).

References

  1. The format of the reference should be unified.

AU: done.

Comments on the Quality of English Language

The English expression throughout the manuscript is very confusing, and the content in the introduction and discussion sections is difficult to understand. The methodology, discussion, and conclusion do not adequately support the purpose of the study. The topic is interesting, but the paper requires a comprehensive revision.

AU: an extensive English revision was made and we used the English editing revision of MDPI service.

We used the MDPI English editing service.

Reviewer 3 Report

Comments and Suggestions for Authors

This study evaluates the applicability and usefulness of the ClassyFarm platform in 25 beef cattle farms of the Marche region. The selected farms have been visited and judged using Classyfarm checklists and comparing the results to national data.

As regards the results, all livestock farms received a lowest rating for biosecurity, even if in line with the national average, and the highest scores for animal welfare.  

The manuscript is attractive for readers not only in the field but also for FBOs and consumers; it's interesting especially considering the role of biosecurity in fighting antimicrobial resistance.

As concerns Materials and Methods, it is unclear how the scores are given (the sum of the parameters found in the checklists maybe?) and how these scores are converted to judgements. Furthermore, on animal welfare, the score corresponding to the judgements (as done in lines 114-116) should be specified. The section "Materials and Methods" should be expanded and clarified with more details.

In addition to this, it is not clear how the total values assigned to the 25 farms for biosecurity and animal welfare Assessment appeared to be identical, not having considered the number of items that make up the two aspects respectively (please see lines 224-228).

Below the list of the grammar mistakes and typographical errors found in the text and points to be clarified:

Line 21: the word “animal” before “biosecurity” is not necessary

Line 43: it would be better to use parentheses instead of square brackets.

Line 51: Please add the preposition “to” after “guarantee”

Lines 53-55: This statement is not exactly true. Joining the ClassyFarm system is on voluntary basis but it becomes a compulsory prerequisite to access CAP (Common agricultural policy) funding (as indicated in the bibliographic reference 5 ). The authors should correct the sentence.

Line 56: It would be better to use the expression “livestock species” instead of “domestic animals”

Lines 66-69: The sentence construction could be improved. e.g. “A recent study of Tamba et al. [11] classified, Italian cattle farms in two types considering consistency and geographic location: the intensive cattle farming in the northern area of the country; the extensive cattle farming in the middle and southern parts of the country.” Please check if the meaning is retained.

Line 80: Please rephrase the sentence like this “little attention, to date, has been paid to…” erasing “until now” and using “to date”.

Lines 96; 101: Do you mean “checklists”? Please explain how many checklists have been used.

Line 104: Please add the word “regarding” before “Biosecurity”. Why authors use just sometimes capital letters for the words “biosecurity “ and “animal welfare”? Authors should explain it and  standardize the way they refer to these aspects throughout the text.

Lines 112-113: Please use the “CFp” acronym in table captions too.

Table 2: Please explain what do you mean with “stockpersons”

Lines 122-123: is this repetition necessary?

Line 125-126: Authors should rephrase the sentence in this way “Bar graphs were used to illustrate how..”

Line 148: Please replace the preposition “for” with “to” before “most of..”

Lines 152-153: Please rephrase the sentence in a more correct way, replacing “for” with “to” “it was given to the other farms with a medium-high frequency”

Lines 153-155: Authors should rephrase the sentence in a more correct way “At the same time, the “excellent” was given to  most of the farms to low-medium frequency  and only 7 farms had a low frequency.”

Lines 156-157: There is a unnecessary “of” after “76”. Please delete it. Authors should rephrase part of the sentence in this way “these low ratings totalled 99”.

Line 159: please replace “overall animal welfare” with the acronym “OAW” introduced before.

Line 165: please replace “have” with “had”

Line 167: Please use the correct form “biosecurity”

Lines 188-190: Please rephrase the sentence like this for major clarity: “Figure 5a shows evidently that  the fattening farms obtained the highest rate in Area C, while the lowest rate in Big Hazards, especially due to the scores of 1-F and 2-F.” Please check if the meaning is retained.

Line 191: “in general they were much higher than the biosecurity one” Please rephrase the sentence for major clarity

Line 201: Do you mean “obtained”? Please check the verb.

Lines 205-206: “During this study, the numerical consistency and variability in breeding techniques of twenty-five beef cattle farms was assessed in the Marche region.” This study rather evaluates how overall animal welfare and biosecurity overall biosecurity parameters ratings vary because of the different breeding techniques and numerical consistency using ClassyFarm checklists. The variability in breeding techniques is not actually described, so the sentence is not correct. Please rephrase it.

Line 208: Please add a bibliographical reference after “ones”

Lines 224-225: Authors should explain the sentence more clearly: “The total values assigned to all the 25 farms for Biosecurity and Animal Welfare Assessment appeared to be identical, not having considered the number of items that make up the two aspects respectively”

Lines 229-234: The same concept is repeated in these two sentences. Please summarize this part.

Line 230: Please add “farms” before “health”

Line 246: Please delete “the” before “animal welfare”

Line 252: Please use the common used form “in the region of Marche”

Line 254: please replace “these aspects” with “the aspects relating animal welfare” for major clarity.

Line 257: Please replace “then” with “consequently” between commas. Please replace “will be” with “are”

An English language revision is necessary. The conclusions are consistent with the evidence and arguments presented.

Comments on the Quality of English Language

An English language revision is necessary. 

Author Response

Comments and Suggestions for Authors

This study evaluates the applicability and usefulness of the ClassyFarm platform in 25 beef cattle farms of the Marche region. The selected farms have been visited and judged using Classyfarm checklists and comparing the results to national data.

As regards the results, all livestock farms received a lowest rating for biosecurity, even if in line with the national average, and the highest scores for animal welfare.

The manuscript is attractive for readers not only in the field but also for FBOs and consumers; it's interesting especially considering the role of biosecurity in fighting antimicrobial resistance.

As concerns Materials and Methods, it is unclear how the scores are given (the sum of the parameters found in the checklists maybe?) and how these scores are converted to judgements. Furthermore, on animal welfare, the score corresponding to the judgements (as done in lines 114-116) should be specified. The section "Materials and Methods" should be expanded and clarified with more details.

In addition to this, it is not clear how the total values assigned to the 25 farms for biosecurity and animal welfare Assessment appeared to be identical, not having considered the number of items that make up the two aspects respectively (please see lines 224-228).

AU: A more detailed explanation of the score is given in the Introduction.

Below the list of the grammar mistakes and typographical errors found in the text and points to be clarified:

Line 21: the word “animal” before “biosecurity” is not necessary

AU: eliminated.

Line 43: it would be better to use parentheses instead of square brackets.

AU: done (line 45).

Line 51: Please add the preposition “to” after “guarantee”

AU: done (line 61).

Lines 53-55: This statement is not exactly true. Joining the ClassyFarm system is on voluntary basis but it becomes a compulsory prerequisite to access CAP (Common agricultural policy) funding (as indicated in the bibliographic reference 5 ). The authors should correct the sentence.

AU: changed (lines 63-67).

Line 56: It would be better to use the expression “livestock species” instead of “domestic animals”

AU: done (lines 67-68).

Lines 66-69: The sentence construction could be improved. e.g. “A recent study of Tamba et al. [11] classified, Italian cattle farms in two types considering consistency and geographic location: the intensive cattle farming in the northern area of the country; the extensive cattle farming in the middle and southern parts of the country.” Please check if the meaning is retained.

AU: done (lines 105-108).

Line 80: Please rephrase the sentence like this “little attention, to date, has been paid to…” erasing “until now” and using “to date”.

AU: eliminated and changed the sentence.

Lines 96; 101: Do you mean “checklists”? Please explain how many checklists have been used.

AU: In Introduction an explanation of the checklist used for beef cattle farms was given.

Line 104: Please add the word “regarding” before “Biosecurity”. Why authors use just sometimes capital letters for the words “biosecurity “ and “animal welfare”? Authors should explain it and  standardize the way they refer to these aspects throughout the text.

AU: done (line 153) and standardized.

Lines 112-113: Please use the “CFp” acronym in table captions too.

AU: done (lines 161-162).

Table 2: Please explain what do you mean with “stockpersons”

AU: changed, done.

Lines 122-123: is this repetition necessary?

AU: eliminated.

Line 125-126: Authors should rephrase the sentence in this way “Bar graphs were used to illustrate how..”

AU: done (lines 172-173).

Line 148: Please replace the preposition “for” with “to” before “most of..”

AU: substituted with “for” by the English editor (line 194).

Lines 152-153: Please rephrase the sentence in a more correct way, replacing “for” with “to” “it was given to the other farms with a medium-high frequency”

AU: done and revised by the English editor (lines 197-199).

Lines 153-155: Authors should rephrase the sentence in a more correct way “At the same time, the “excellent” was given to  most of the farms to low-medium frequency  and only 7 farms had a low frequency.”

AU: done and revised by the English editor (lines 199-201).

Lines 156-157: There is a unnecessary “of” after “76”. Please delete it. Authors should rephrase part of the sentence in this way “these low ratings totalled 99”.

AU: done and revised by the English editor (lines 202-203).

Line 159: please replace “overall animal welfare” with the acronym “OAW” introduced before.

AU: done (line 206).

Line 165: please replace “have” with “had”

AU: done (line 210).

Line 167: Please use the correct form “biosecurity”

AU: done (line 212).

Lines 188-190: Please rephrase the sentence like this for major clarity: “Figure 5a shows evidently that  the fattening farms obtained the highest rate in Area C, while the lowest rate in Big Hazards, especially due to the scores of 1-F and 2-F.” Please check if the meaning is retained.

AU: done and revised by the English editor (lines 218-220).

Line 191: “in general they were much higher than the biosecurity one” Please rephrase the sentence for major clarity

AU: done and revised by the English editor (lines 221-223).

Line 201: Do you mean “obtained”? Please check the verb.

AU: done (line 232).

Lines 205-206: “During this study, the numerical consistency and variability in breeding techniques of twenty-five beef cattle farms was assessed in the Marche region.” This study rather evaluates how overall animal welfare and biosecurity overall biosecurity parameters ratings vary because of the different breeding techniques and numerical consistency using ClassyFarm checklists. The variability in breeding techniques is not actually described, so the sentence is not correct. Please rephrase it.

AU: done (lines 236-244).

Line 208: Please add a bibliographical reference after “ones”

AU: done (line 244).

Lines 224-225: Authors should explain the sentence more clearly: “The total values assigned to all the 25 farms for Biosecurity and Animal Welfare Assessment appeared to be identical, not having considered the number of items that make up the two aspects respectively”

AU: the sentence has been deleted.

Lines 229-234: The same concept is repeated in these two sentences. Please summarize this part.

AU: done and shifted in the introduction part (lines 96-104).

Line 230: Please add “farms” before “health”

AU: the sentence has been changed.

Line 246: Please delete “the” before “animal welfare”

AU: done (line 284).

Line 252: Please use the common used form “in the region of Marche”

AU: changed by the English editor (line 289).

Line 254: please replace “these aspects” with “the aspects relating animal welfare” for major clarity.

AU: done (lines 291-292).

Line 257: Please replace “then” with “consequently” between commas. Please replace “will be” with “are”

AU: done (lines 294,297).

An English language revision is necessary.

AU: done by the MDPI English editing service.

The conclusions are consistent with the evidence and arguments presented.

Comments on the Quality of English Language

An English language revision is necessary.

AU: done by the MDPI English editing service.

Reviewer 4 Report

Comments and Suggestions for Authors

Dear Authors,

Thank you for addressing Biosecurity in beef cattle. I appreciate your attempt to investigate indicators and the use of a checklist, however, I am not sure if this manuscript can be published in the present form.

Please find attached my comments.

1.       The title should be revised (A survey ON biosecurity)

2.       L 21 and L32: what do you mean with “even if”

3.       L17: Can you confirm the research question/s please? Are you investigating if this system can be adopted to assess animal welfare/health in this region or are you reporting the findings of the official controls?

4.       L20 I do not think that the Veterinary Services would “judge” farmers… maybe they perform an official control.

5.       L34 Can you provide evidence about and discuss the reason why non-intensive breeding systems and small farms should be better (“characterized by particular attention to animal welfare”) than other systems?

6.       L47 Could you please provide a better description of the services provided by “Vetinfo”? The definition “public veterinary services” can be misleading. Maybe you would like to say that stakeholders can access to several documents, useful links etc?

7.       L 51 Animal welfare and food safety are relevant not only for consumers. Could you please improve this part?

8.       L59 When I read this part, I ask myself if everybody (even not-trained persons) can assess biosecurity/welfare at farm level with the checklists.

I am sure that also the veterinarians had to be trained before using the checklists. Could you provide an insight about that? I also believe that the checklists are useful instruments that support the official veterinarians and help standardizing the “list” of items that should be checked during a control. This is important also to make sure that all the farmers are treated in the same way… Can you discuss these points?

9.       L 63: Could you please describe the indicators to better understand?

10.   L66-71 You provide information about the size of the farms, the breeds, and the origin of the calves. Which is the purpose of this paragraph?

11.   L 73 The papers were based in one Province (South Tyrol)

12.   L83-89 Here you describe why the breeding system in the Region Marche is unique. However, are you sure that this traditional rural system is not comparable to other systems? How is the situation in the neighboring regions?

13.   Back to the point n. 3. Are you investigating if CF is applicable and/or useful? How can you answer these questions?

14.   Are the data publicly available or did you get an informed consent from the farmers that maybe can be identified on the map? How did you make sure that their privacy could be protected? Did you require an approval of the local ethical commission and if not why?

15.   If you think that the data on table 4 can provide relevant information you could maybe  display them using a graph otherwise you can shift the table to the additional material

16.   RESULTS: it is difficult to read the description of the results. Could you please improve this part? A brief and easy description of the main findings would help the readers.

17.   L205-208: The data that confirm that the farms are small in average is important, but not a highlight. As for the results, I would think about the possibility of summarizing this information in a short and effective sentence.

18.   L 209: of course, there are strengths and areas of improvement. Please discuss them thoroughly. If the available studies focus only on dairy farms, discuss the main differences and similarities.

19.   L211-214: In general, a definition of Biosecurity and the description of the most important biosecurity measures that can be implemented are missing.

The implementation of effective biosecurity measures is important to reduce the transmission of diseases and in turn helps to contrast AMR. Apparently, the topic of this paper is not AMR. However, if you want to mention it, I recommend you to write a paragraph explaining very well these topics together. Maybe you can use the concepts written in the paragraph L229-238? However, this could be part of the introduction, to better explain the importance of implementing a risk-based systems that facilitate the official controls. As you do not assess the use of antibiotics (there should be a specific checklist about that), I would suggest staying focused on biosecurity and welfare.

20.   L216: If you want to describe the strengths of the farmers, I recommend doing it very well now. What does it mean that they are “careful”? Any other strength?? I still do not understand why they are so good at preserving animal welfare and at the same time why the scores are lower for biosecurity. Please discuss the most relevant items (for instance: do they inspect daily the animals? Do they apply a quarantine? If not why, how can they improve?)

21.   Could you please better describe how your findings support the statement “In addition, it was possible to outline not only the usefulness of the risk-based assessment provided by the CFp, but also the possibility of correlating the improvements in herd health management with production”

22.   L219- 228 Here you provide information about the weakness of CF. Please expand this paragraph

23.   Conclusion: Unfortunately, this manuscript does not provide thorough information about the CF. How can you prove that it is reliable?

In conclusion, I do agree when you state that “an accurate analysis of these results should stimulate farmers and veterinarians to clearly identify critical points and then improve farming management”. Please do this analysis, discuss the results, and then disseminate the findings among vets and farmers. It can very useful.

Comments on the Quality of English Language

English language check required? 

Author Response

Comments and Suggestions for Authors

Dear Authors,

Thank you for addressing Biosecurity in beef cattle. I appreciate your attempt to investigate indicators and the use of a checklist, however, I am not sure if this manuscript can be published in the present form.

Please find attached my comments.

  1. The title should be revised (A survey ON biosecurity)

AU: done.

  1. L 21 and L32: what do you mean with “even if”

AU: we changed.

  1. L17: Can you confirm the research question/s please? Are you investigating if this system can be adopted to assess animal welfare/health in this region or are you reporting the findings of the official controls?

AU: We are describing how the system is adopted to assess animal welfare/health in Marche region and discuss the results.

  1. L20 I do not think that the Veterinary Services would “judge” farmers… maybe they perform an official control.

AU: We changed it.

  1. L34 Can you provide evidence about and discuss the reason why non-intensive breeding systems and small farms should be better (“characterized by particular attention to animal welfare”) than other systems?

AU: We changed it (lines 36-37). The family breeding with very few heads can lead the farmed to be more in contact with the animals and to report quicker some diseases, since the animal observation measures can be used very often during the daily routines in the farm.

  1. L47 Could you please provide a better description of the services provided by “Vetinfo”? The definition “public veterinary services” can be misleading. Maybe you would like to say that stakeholders can access to several documents, useful links etc?

AU: done (lines 52-59).

  1. L 51 Animal welfare and food safety are relevant not only for consumers. Could you please improve this part?

AU: done (lines 59-63).

  1. L59 When I read this part, I ask myself if everybody (even not-trained persons) can assess biosecurity/welfare at farm level with the checklists.

I am sure that also the veterinarians had to be trained before using the checklists. Could you provide an insight about that? I also believe that the checklists are useful instruments that support the official veterinarians and help standardizing the “list” of items that should be checked during a control. This is important also to make sure that all the farmers are treated in the same way… Can you discuss these points?

AU: the checklist and relative use are explained in Introduction.

  1. L 63: Could you please describe the indicators to better understand?

AU: the reference used [9] (lines 91-93) can show in detail which are the behavioural indicators taken into account.

  1. L66-71 You provide information about the size of the farms, the breeds, and the origin of the calves. Which is the purpose of this paragraph?

AU: the purpose of this paragraph (lines 105-111) is to better explain the farming conditions in Italy.

  1. L 73 The papers were based in one Province (South Tyrol)

AU: done (line 116).

  1. L83-89 Here you describe why the breeding system in the Region Marche is unique. However, are you sure that this traditional rural system is not comparable to other systems? How is the situation in the neighboring regions?

AU: this topic was revised and better described also including the earthquake event of year 2016 (lines 123-137).

  1. Back to the point n. 3. Are you investigating if CF is applicable and/or useful? How can you answer these questions?

AU: We answer in an affirmative way after the revisions added on the description of CF application are reported in Introduction and Discussion.

  1. Are the data publicly available or did you get an informed consent from the farmers that maybe can be identified on the map? How did you make sure that their privacy could be protected? Did you require an approval of the local ethical commission and if not why?

AU: thank you for considering this point. The spots we have created on the map are so large that they do not allow the geographical location of the companies, precisely to protect their privacy. the map is only useful to indicate to the reader an overview of the spatial diffusion of the farms. In accordance with the Italian (Legislative Decree 26/2014) and European (EU Regulation 63/2010) regulations, an ethical permit is not necessary to work on these animals since we do not carry out any manual work on them.

  1. If you think that the data on table 4 can provide relevant information you could maybe display them using a graph otherwise you can shift the table to the additional material

AU: done, we changed the table into histograms.

  1. RESULTS: it is difficult to read the description of the results. Could you please improve this part? A brief and easy description of the main findings would help the readers.

AU:  the Results section has been revised to make it be clearer and easy to understand, even with the help of the English editing revisions.

  1. L205-208: The data that confirm that the farms are small in average is important, but not a highlight. As for the results, I would think about the possibility of summarizing this information in a short and effective sentence.

AU: done (lines 242-244).

  1. L 209: of course, there are strengths and areas of improvement. Please discuss them thoroughly. If the available studies focus only on dairy farms, discuss the main differences and similarities

AU: this sentence is removed because not related to data of our study.

  1. L211-214: In general, a definition of Biosecurity and the description of the most important biosecurity measures that can be implemented are missing.

AU: a larger description of the Biosecurity measures was written in the Discussion section (lines 245-276).

The implementation of effective biosecurity measures is important to reduce the transmission of diseases and in turn helps to contrast AMR. Apparently, the topic of this paper is not AMR. However, if you want to mention it, I recommend you to write a paragraph explaining very well these topics together. Maybe you can use the concepts written in the paragraph L229-238? However, this could be part of the introduction, to better explain the importance of implementing a risk-based systems that facilitate the official controls. As you do not assess the use of antibiotics (there should be a specific checklist about that), I would suggest staying focused on biosecurity and welfare.

AU: we moved the paragraph in the introduction section just mentioning the topic of AMR (lines 96-104).

  1. L216: If you want to describe the strengths of the farmers, I recommend doing it very well now. What does it mean that they are “careful”? Any other strength?? I still do not understand why they are so good at preserving animal welfare and at the same time why the scores are lower for biosecurity. Please discuss the most relevant items (for instance: do they inspect daily the animals? Do they apply a quarantine? If not why, how can they improve?)

AU: we added some other information derived by the results, that can be individually checked in the supplementary materials.

  1. Could you please better describe how your findings support the statement “In addition, it was possible to outline not only the usefulness of the risk-based assessment provided by the CFp, but also the possibility of correlating the improvements in herd health management with production”

AU: it was better described.

  1. L219- 228 Here you provide information about the weakness of CF. Please expand this paragraph

AU: this sentence was removed.

  1. Conclusion: Unfortunately, this manuscript does not provide thorough information about the CF. How can you prove that it is reliable?

AU: done.

In conclusion, I do agree when you state that “an accurate analysis of these results should stimulate farmers and veterinarians to clearly identify critical points and then improve farming management”. Please do this analysis, discuss the results, and then disseminate the findings among vets and farmers. It can very useful.

AU: this part was added in the conclusion.

We used the MDPI English editing service.

Round 2

Reviewer 2 Report

Comments and Suggestions for Authors

This manuscript aims to analyze the applicability of CFp in 25 beef cattle farms in the Marche Region, central Italy.   CFp was activated to assess livestock farms' biosecurity and welfare and orovides information to the consumers.   All livestock farms received the lowest rating for animal biosecurity, even if in line with the national average and the highest scores for animal welfare.   This result reflects the breeding typical of this region, characterized by particular attention to animal welfare, guaranteed by non-intensive and family-run breeding.   The article was deemed worthy of publication after revision.

Author Response

Dear reviewer,

thanks a lot for your revisions and your decision to assess that the article is ready for publication.

Kind regards.

Reviewer 4 Report

Comments and Suggestions for Authors

Dear Authors,

 I appreciate your attempt to review the paper, however, I still do not think that this manuscript can be published 

Please find attached my comments.

1.       “The family breeding with very few heads can lead the farmed to be more in contact with the animals and to report quicker some diseases, since the animal observation measures can be used very often during the daily routines in the farm”: please provide evidence about this opinion

2.       The papers were based in one Province (South Tyrol) = Alto Adige

3.       The discussion is still to poor and not all the sentences are supported by relevant literature and/or discussed together with the findings (L247-265)

I I am afraid 

Author Response

Thanks a lot for your comments.

We added the new parts in yellow and highlighted and the revision tracking of the modified references.

  1. “The family breeding with very few heads can lead the farmed to be more in contact with the animals and to report quicker some diseases, since the animal observation measures can be used very often during the daily routines in the farm”: please provide evidence about this opinion
    AU: done (line 304-305).
  2. The papers were based in one Province (South Tyrol) = Alto Adige
    AU: we modified it (line 116).
  3. The discussion is still to poor and not all the sentences are supported by relevant literature and/or discussed together with the findings (L247-265)
    AU: we modified and amplified the discussion with other references.

Hoping the article can be considered ready for the publication, 
We thank you in advance.